# Designing Peer-to-Peer Systems as Closed Knowledge Commons

Erick Lavoie
erick.lavoie@unibas.ch
University of Basel, Basel, Switzerland

## Abstract

In contrast to many online services based on client-server infrastructure, peer-to-peer systems are usually designed as open commons. This is partly because, by design, peer-to-peer systems replicate data on end-user devices and typically use open implementations, precluding access control. Open commons however lower incentives for end users to contribute the resources necessary to cover development and maintenance costs, resulting in chronic underfunding and few offerings of mature peer-to-peer alternatives.

In this paper, we show how to design peer-to-peer systems as closed commons by making the replication of updates conditional to proven contributions, tracked by a blockchain or eventually-consistent ledger. We also present an economic model that incentivizes users to support both developers of the system and content producers. We finally identify factors that suggest our economic model might be cost-competitive with cloud-hosting for compatible applications.

***CCS Concepts:*** • **Computer systems organization → Peer-to-peer architectures**.

***Keywords:*** economics, peer-to-peer, append-only logs

**ACM Reference Format:**
Erick Lavoie. 2023. Designing Peer-to-Peer Systems as Closed Knowledge Commons. In *4th International Workshop on Distributed Infrastructure for the Common Good (DICG '23), December 11–15, 2023, Bologna, Italy*. ACM, New York, NY, USA, 6 pages. https://doi.org/10.1145/3631310.3633491

## 1 Introduction

Over the last decade, open source communities have pioneered many initiatives to redecentralize major Internet services, often using peer-to-peer architectures [4, 17, 24]. However, the economic models used by many peer-to-peer projects are not sufficient to cover their development and maintenance costs and most projects are effectively subsidized through other means, e.g. by volunteers offering their work freely and a small proportion of users donating. The situation is partly caused by the technical workings of the peer-to-peer protocols themselves: replicating data on every user devices precludes the ability of controlling access to data after it has been replicated. Peer-to-peer projects then simply eagerly replicate data to whoever requests it, effectively implementing an open data commons.

Client-server applications, in contrast, are often designed with a clear and closed boundary: access to the server is conditional on

having previously paid for the services. Users therefore have a clear incentive to contribute. The much larger number of companies using closed client-server models compared to peer-to-peer alternatives suggests the former currently better succeeds at covering development and maintenance costs.

In this paper, we show that a *closed commons* for peer-to-peer systems is not only possible but may also be cheaper to operate than client-server applications hosted in clouds. We first present an economic model to support developers of core protocols and content producers. Our model is voluntary, with producers of content able to make some of their content freely available or contribution-conditional, and contributors able to choose which producers they want to support. We then make our economic model practical with the design of a replication primitive that is conditional on contributions. Our primitive is based on the conjunction of two key insights: 1) updates of developer and user generated code and content can be restricted to contributors of time and/or financial resources tracked by producer-specific tokens, and 2) contributors have a natural incentive to only replicate updates with other contributors. We finally identify economic and technical factors that are likely to make peer-to-peer hosted applications based on our economic model competitive with cloud-hosted offerings.

The rest of this paper is structured as follows: we present our economic model in Section 2, we describe our contribution-conditional replication primitive in Section 3, we identify favorable economic and technical factors in Section 4, we relate our work to similar fields in Section 5, and we conclude with potential future research directions in Section 6.

## 2 Economic Model

In this section, we present an economic model for peer-to-peer systems that forms a *voluntary closed commons* of knowledge production with the ability to remunerate producers with contributions from consumers. Our economic model is based on *voluntary participation*, which we define as follows:

**Voluntary Participation**: *Producers can optionally choose to ask for contributions in exchange for their updates. Contributors choose who they support, but only get updates from producers for whom they meet contribution requirements.*

We now present the participating economic actors, exchange flows, examples of possible contribution schemes, and finally show that the actors have incentives to maintain the commons as a voluntary but closed system, contributing to the long-term sustainability of an implementation.

### 2.1 Actors

Our model has two kinds of actors:

- **Producers**: Produce knowledge in the form of updates, e.g. software updates or content, for contributors in exchange for tokens.

- **Contributors**: Obtain updates in exchange for tokens. Provide computing resources necessary for replication (storage, network bandwidth, and computing power). Actively maintain the commons boundary by reserving updates to other contributors that meet the contribution requirements of those updates.

*Producers* are similar to knowledge producers, such as software developers, bloggers, writers, video producers, etc. Similar to writers on major blogging platforms, they can ask for tokens and reserve some updates only for those who paid them. However, in contrast to many current online platforms, the developers that maintain the platform do not have a privileged position, e.g. that allows them to charge a percentage of other transactions happening on the platform. They can only request contributions for the updates they themselves provide. Moreover, users of the platform voluntarily choose which updates they will use and therefore retribute with tokens.

*Contributors* are in some aspects similar to regular consumers of content: they obtain updates that they can use an unlimited number of times. For example, once they have obtained a software update, they can continue using the updated software in perpetuity. Similarly, they can read blog posts or watch videos an unlimited number of times. However, we explicitly call them contributors because they participate much more actively than is normally expected of consumers in other systems: 1) they participate in the replication of updates with their device; 2) they provide the necessary computing resources for replication, including storage required to replicate all updates, network bandwidth to disseminate the updates, and computation for providing services; and 3) they actively enforce the commons boundary by only sending updates to contributors that have met the producer requirements (as they themselves did).

## 2.2 Exchange Flows

For both producers and contributors, exchanges are structured around *updates* because peer-to-peer systems based on open protocols and formats cannot limit later accesses to data once it has been replicated. Therefore, the exchange of knowledge for tokens is instead enforced at the moment the knowledge is first replicated, i.e. when a potential contributor requests the update. Trading of updates for tokens serves two purposes with opposite directions of flow.

In one direction, which we call *push*, contributors ask others to propagate their latest updates through the system. We tie a contributor's pushing ability to having contributed at least the minimum requested by developers for software updates. However, we do not compensate the use of computing resources of other contributors partly to avoid the complexity of creating a resource market but mostly because the dominant cost in a peer-to-peer system is developer's time.

In the other direction, which we call *pull*, contributors request the newest updates of producers, in the form of software or content, from other contributors. The contribution requirements in this case are established by the producers but enforced by all contributors. We make a special case of software updates from developers of the system: the same contribution gives both the ability to push a contributor own's updates and pull the latest developer updates.

## 2.3 Possible Contribution Schemes

Multiple schemes of contribution requirements are possible in our model. So far, we have identified two major categories that we describe as follows.

**Time-based subscription**: This scheme is analogous to existing newspaper or blogging platform funding models in which subscribers pay for a time-limited access to the content, e.g. access to all subscriber-limited articles for a one month period, but applied on updates instead. Contributors first obtain tokens that are accepted by the producer, then transfer the tokens to the producer. If using a blockchain, the subscription time starts when the block containing the transaction was added on the blockchain, i.e. validated and possibly mined by validator nodes. If using an eventually-consistent ledger, such as GOC-Ledger [12], the subscription starts when the reception of tokens was acknowledged by the producer. In both cases, determining the time elapsed since the subscription began is performed by another contributor providing updates and is outside the control of the subscriber. A subscriber then will receive all updates that have previously happened at the time of a request, if the request is performed before the subscription expires.

**Update-based subscription**: This scheme uses the publishing of updates as logical events to determine whether a subscriber's request will be fulfilled. Updates are sequentially published in an append-only log, e.g. as 2P-BFT-Log [11], tied to the producer identifier. A producer may then, for example, ask for "one token per update": i.e. if a subscriber has transferred $x$ tokens then they will receive the first $x$ updates from a producer. This scheme also allows discounting earlier updates based on the total number of updates replicated from a producer, e.g. the first fifty updates may be available at half price once at least one hundred updates have been published. In contrast to the time-based scheme, this enables a producer to better distribute costs across subscribers over time and removes the dependency on an external clock.

## 2.4 Incentives

The incentives of both producers and contributors are aligned with sustaining the economic model.

Both software producers that maintain the system as well as content producers have an incentive to maintain the model because they are remunerated from it. They might be tempted to require larger contributions but since their total revenues are equal to the amounts requested times the total number of contributors, they are incentivized to limit the requested amounts so the updates will be accessible to a sufficiently large number of contributors [19]. Producers are however not in a position to force contributors to use their updates, therefore they have to produce updates that are desirable by contributors.

Contributors have an incentive to use a peer-to-peer system because it can potentially provide more affordable replication services than cloud-based alternatives (see Section 4), therefore they can obtain updates from their favorite content providers for less than on other alternatives. They also have an incentive to use an update-based system because they won't have restrictions on how many times they can use the resulting software or content. Finally, contributors are responsible for maintaining the ability of producers to obtain a sufficient livelihood by not providing updates to non-contributors. We expect this mutual dependency between the

developers of the system, the content providers, and contributors will contribute to developing strong loyalty.

Nonetheless, our model does not prevent some participants to freely distribute contribution-restricted updates. This is less of an issue than it may first appear because the vast majority of contributors do not have the technical skills nor the time to modify implementations to circumvent contribution checks. For the sake of argument, let's still assume that among the small number of contributors that have both the skills and time to do so, some do decide to maintain alternative implementations that do not implement checks for contributions while remaining compatible with the rest of the functionalities. They therefore would become non-loyal producers of updates and users of their software updates would become non-contributors (free riders). Non-loyal producers have to maintain alternative implementations while being compensated less than the loyal developers and content providers (potentially not at all), because otherwise the non-contributors would be better off by simply contributing to the loyal producers to get the updates legitimately. This therefore puts non-loyal producers at a disadvantage, for otherwise if they have the required skills to contribute to development they might as well contribute legitimately and be compensated through the official channels. Moreover, non-contributors still have to trust non-loyal producers to not use the opportunity to create malicious software that could e.g. steal tokens, use computing resources for illegitimate purposes, or steal private data to increase their gains. It seems more than likely that a sufficient majority of contributors, given reasonably priced updates, will prefer getting their updates legitimately to obtain official support and good security and privacy guarantees. Nonetheless, since our goal is to enable producers to cover their costs of production, some non-loyal behaviour can still be tolerated as long as it is sufficiently limited to not threaten the livelihood of loyal producers.

That being said, the possibility of producers becoming non-loyal should not be entirely prevented because it also threatens the loyal developers and content providers with the possibility of starting a competing system if the needs and desires of contributors are not sufficiently taken into account, therefore helping to establish a good balance between the interests of all actors.

## 3 Contribution-Conditional Replication

In this section, we present a replication primitive that ensures the requested amount of tokens by producers has been previously transferred prior to replicating updates. This replication primitive is the core technical enabler of our economic model (Sec. 2).

### 3.1 System Model

Our replication primitive is intended for eventually-consistent replicated databases that contain updates from contributors and whose replicas periodically reconcile their state with other replicas. Our work is based on Git [25], but other systems such as Secure-Scuttlebutt [9, 24] or Hypercore [8, 17] could also be used.

Users might be non-loyal, in which case they might create alternative replicas that do not implement our replication primitive, and therefore do not honour producer requirements, but still propagate updates. However, we assume they have limited resources to do so and will eventually be exposed and blocked by loyal contributors, and therefore the majority of users will use replicas maintained by loyal developers (as discussed in Section 2.4).

We assume every user, i.e. producer or contributor, has a public-private key pair whose public part is used as an identity and the private part is used to sign messages for authenticating their provenance. We assume *loyal* users do not share private keys, therefore signatures uniquely identify them. We do not assume anything about non-loyal users.

### 3.2 Token Layer

Users use the same public identity to exchange tokens with other users, e.g. through a blockchain platform such as Bitcoin or Ethereum or using an eventually-consistent ledger, such as GOC-Ledger [12]. In the first case, double-spending is prevented by the platform, while in the second case it will be detected after the fact, e.g. by using 2P-BFT-Log [11]. In case of detection, 1) loyal users will provide negative exposure of offending users; 2) the correct replicas operated by loyal users will eventually replicate the proof that a fork happened [11] (that enabled double-spending [12]) and that proof will be used to forever prevent the offending users from ever participating in replication with loyal users again.

Each producer emits their own tokens, e.g. implementing local crypto-tokens [13], that can be bought e.g. in exchange for other producer tokens, through public exchanges, or simply through a website with a payment mechanism.

### 3.3 Update Layer

Independently of the layer used for token exchanges, users publish their updates in an append-only log that sequentially orders all updates and is eventually-consistent even in the presence of malicious users, such as 2P-BFT-Log [11]. This is necessary to disambiguate which update is paid for by which tokens. If users break the sequentiality of their updates, they are eventually automatically blacklisted.

Producers and contributors may have as many public logs, i.e. logs without contribution requirements, as they fancy but they need to contribute tokens for each individual log to be replicated by the system. This prevents Sybil attacks and decreases the appeal of the system for trolls, by associating an economic cost to each replicated log.

The first message of a log describes the contribution requirements for obtaining all subsequent updates. Changes in contribution requirements may be implemented by terminating the current log with a message pointing to a new log listing different contribution requirements, with contributors having to explicitly decide to subscribe to the new log and ensure they meet new requirements, before receiving the new updates.

Updates are encoded as Git commits [6] and signed by the producer under a self-certifying branch reference [11]. Apart from the contribution checks introduced by our replication primitive in the next section (Sec. 3.4), they are replicated using the standard Git replication protocol [6].

### 3.4 Replication Primitive

Algorithm. 1 lists the three operations that form our replication primitive: all three operations are analoguous to the push and pull operations of Git but specialized for updates on append-only logs and token contributions. Operations are event-driven and their implementation describes how a replica reacts to requests from other replicas. These operations are the only interface through which a replica may obtain updates from another replica. For all operations,

the user performing the request, i.e. the *requester*, must be authenticated first, e.g. using SSH or the secret handshake protocol [23], otherwise the operation aborts immediately.

---

**Algorithm 1** Contribution-Conditional Replication

---

 1: **Require**:
  • *devId*: identifier receiving tokens for development
 2: *producers* ← { *devId* }                     ▷ Identifiers actively replicated
 3: *contributions* ← *empty ledger replica*
 4:
 5: **upon** PUSH-UPDATES(*requester*, *frontier*)
 6:    **Preconditions:** *requester* is authenticated and *frontier* is valid.
 7:      **if** *requester* ∈ *producers* **then**
 8:          **store** updates from *requester* up to *frontier*
 9:
10: **upon** PULL-UPDATES(*requester*, *producer*, *frontier*)
11:    **Preconditions:** *requester* is authenticated, *producer* is known, and *frontier* is valid.
12:      **if** *contributions* from *requester* meet requirements from *producer* **then**
13:          **send** updates from *producer* to *requester* newer than *frontier*
14:
15: **upon** PUSH-CONTRIBUTIONS(*requester*, *contributions*')
16:    **Preconditions:** *requester* is authenticated and *contributions*' is valid.
17:      **if** *requester* ∉ *producers* **and** *contributions*' satisfy requirements of developers for *requester* **then**
18:          **merge** *contributions* with *contributions*'
19:      **else if** *requester* ∈ *producers* **then**
20:          **merge** *contributions* with *contributions*'
21:      **for each** *contributor* ∈ *contributions* **do**
22:          **if** *contributions* from *contributor* satisfy requirements from developers **then**
23:              *producers* ← *producers* ∪ { *contributor* }

---

Before execution, a replica needs *devId*, the identifier of the account from which developers of the system are compensated. This identifier is distributed along or inside the embedding peer-to-peer application.

When a replica is initialized for the first time, two variables are initialized. The set of *producers* only contains *devId*, i.e. the identifier of the developers of the application. The ledger replica containing *contributions* is empty. Both are persisted between executions.

The first operation, PUSH-UPDATES, enables a *requester*, as identified by their public key, to push their updates to another replica. In addition to authenticating the requester, the *frontier*, which represents the latest state of *requester*'s log, must be valid, i.e. it must be correct according to the expected format of updates and invariants of the log. If neither of these conditions is met, the operation aborts immediately. Otherwise, only if *requester* is in the current set of producers, do their updates are stored and propagated further.

The second operation, PULL-UPDATES is similar to PUSH-UPDATES but has updates flowing in the other direction. It enables a *requester*, also identified by their public key, to pull updates from a producer, also identified by their public key, from another replica. In addition

to authenticating the *requester*, the producer must be known, and the frontier, i.e. the latest state of the producer log known by the requester, must be valid otherwise the operation aborts immediately. If previous contributions from *requester* intended for *producer* are sufficient, then updates more recent than *frontier* are sent back to *requester*. Otherwise, no updates are sent.

The third operation, PUSH-CONTRIBUTIONS, is a separate operation from PUSH-UPDATES because otherwise the preconditions of the latter would prevent updates representing tokens transfers from propagating. In addition to an authenticated *requester*, it receives *contributions*', a set of contributions represented as a state-based CRDT (e.g. a grow-only set of references to blocks of a blockchain or the state of an eventually-consistent ledger, such as GOC-Ledger [12]). Assuming *contributions*' is valid, then it is merged with *contributions* inside the local replica under two conditions: either 1) *requester* is not already in the *producers* set and *contributions*' satisfies the requirement from developers to make *requester* a producer as well; or 2) *requester* is already a producer in which case it is allowed to propagate their contributions and those of others.

## 4 Economics of Cloud Hosting compared to Peer-to-Peer Hosting

Developing and maintaining cloud- and peer-hosted applications entails a variety of costs including but not limited to hardware procurement and replacement, Internet access, developer and maintainer time, and energy supply. We highlight hereafter cases in which the economics work differently between both and suggest peer-to-peer hosting can be economically competitive.

Cloud hosting costs include replacement of hardware every 5-6 years [2, 3, 15], as well as cooling costs, which represent as much as 30% of the power consumption of a data center [30]. In contrast, peer-to-peer applications leverage spare resources from end user devices that have otherwise been paid for other usages. Because end user devices are widely distributed geographically, they do not require additional cooling infrastructure. In addition, from an end-user perspective, cloud hosting *always involves more and more expensive hardware* than peer-to-peer hosting because a client device is needed in both cases but servers with strong capabilities are only needed for cloud-hosting.

In cloud hosting, network traffic costs are typically proportional to the total amount of data downloaded from a data centre but uploading is typically free [1]. In contrast, many end-users have home data plans with fixed monthly prices and unlimited data both in download and upload (but bandwidth throttling) (ex: [22]). Therefore, data transfer costs, when performed directly between end-users, are going to be equal or less than data transfers to and from cloud-hosted applications. In effect, peer-to-peer transfers leverage spare data transfer capacity that end users are already paying to their Internet Service Provider.

In a cloud, the baseline power requirements to maintain constant availability are significant and force cloud operators to require instantly dispatchable energy sources and/or significant energy storage. In both cases, these are more expensive than intermittent renewable sources used directly at the production site [29]. This is because cloud operators cannot control when users will actually use their service. In contrast, mobile devices are already equipped

with batteries and end-users have the choice to defer their energy-intensive computing activities to when energy is the cheapest, which typically will occur during periods of abundant sunlight or strong winds. Moreover, since end-users have visibility of their monthly energy consumption and an economic incentive to defer usage to periods of lower prices, peer-to-peer systems should more easily adapt to renewable energy availability, without the need of artificial incentives, nudging, or extra monitoring infrastructure.

In addition to the intrinsic scalability of peer-to-peer systems, because more users contribute more computing resources, these trends suggest that the platform costs of running peer-to-peer systems will be lower than those of cloud hosting. Assuming all other costs being similar, e.g. remunerating developers and maintainers as well as content producers, this means peer-to-peer hosting may cost less, leading potentially to more affordable services.

## 5 Related Work

To the best of our knowledge, we are the first to propose a contribution-conditional replication primitive for eventually-consistent peer-to-peer systems, with an associated economic model and economic analysis. That being said, other work has addressed similar themes in other contexts, which we briefly survey.

### 5.1 Economics of Distributed Computing

In 1998, Shapiro and Varian [19] conceptualized unique properties of digital products and services distributed over the Internet, which they had called the *network economy*. Their analysis, among other topics, explained how to maximize the revenues and profit margins of technology companies using new marketing strategies that were not common or even possible in other economic sectors, with wide and successful adoption. Varian later became Chief Economist at Google and has continued to write on the topic [26, 27].

In 2009, researchers at the RadLab at UC Berkeley [7] described the underlying economic and technical factors, including elasticity and risk shifting, that clouds made possible and described the economies of scale achievable by moving on-premise infrastructure into the cloud. Their analysis successfully foreshadowed the growth of the startup ecosystem based on cloud hosting. Over a decade later, clouds appear to not remain cost competitive for large and mature companies, with companies such as Dropbox bringing back their computing infrastructure in-house [28].

The previous analyzes have not considered the possibility of organizing computing resources as commons and did not compare the costs of peer-to-peer hosting to those of clouds.

### 5.2 Eventually-Consistent Peer-to-Peer Systems

In parallel, there was intense research on structured peer-to-peer network overlays, e.g. online distributed hash tables [14, 18, 21, 31]. However, as the cloud emerged as a dominant paradigm, general academic interest on peer-to-peer systems waned. A newer generation of systems appeared in the 2010s [8, 17, 24], that could work with intermittent connectivity because they were inspired by decentralized version control systems [25] and therefore *strongly eventually consistent* [20]. This newer paradigm for application development, spearheaded by open-source developer communities, was

fully articulated as *local-first software* [10]. However, eventually-consistent peer-to-peer system designers did not consider the integration of economic layers and designed them as open commons instead.

### 5.3 Blockchains and Peer-to-Peer Accounting

Bitcoin [16] publicly emerged in 2009 as the first peer-to-peer currency and payment system that required no trust in a central authority for creating accounts, emitting currency, or performing transactions. Ethereum [5] followed in 2014 and generalized the system to turing-complete state-machine replication, i.e. execution of general-purpose code, enabling a plethora of new tokens and financial services. However, both platforms require *global consensus* and maintain a *single global ledger*, requiring billions of dollars of investments either in mining hardware or staked capital [13]. This entails transaction fees that are too high for applications in local economics [13], in which participants that know each others repeatedly transact together. An alternative approach is to link token creation to producers, detect double-spending after the fact, and exclude malicious participants [13].

## 6 Conclusion and Future Work

We have presented an economic model for peer-to-peer systems based on *closed commons*. This model is voluntary: producers have the possibility to not require contributions and contributors have the choice of whom they want to support. We have designed a *contribution-conditional* replication primitive as the core operation that enables this economic model, by allowing updates to flow only if contributions were made. We have finally identified economic and technical factors that suggest peer-to-peer hosting based on our economic model might have lower cost of operations than cloud hosting, which highlights an economic niche in which peer-to-peer systems based on closed commons might thrive.

Our economic model could be refined with additional specialized roles, e.g. data transmitter in community networks or hoster on highly-available replicas, beyond those of software update and content producers. Each additional role would extend the replication primitive checks to ensure sufficient contributions have been made to obtain the associated services, prior to replicating updates. Similar to those of other producers, the contribution requirements of additional roles can also be encoded into logs and replicated like any others.

We intend to demonstrate the practicality of both the base economic model as well as possible extensions within software development communities. The aim will be to test the technical ideas and the economic assumptions we have made in this paper, then publish the resulting positive and negative findings. Developers and content producers interested in participating are invited to contact us at the email address mentioned on the front page.

## Acknowledgments

We would like to thank Aljoscha Meyer, Osman Biçer, Jannick Heisch and members of the Secure-Scuttlebutt community for constructive feedback on a draft of this paper. We would also like to thank Prof. Christian Tschudin for providing a stimulating research environment and having allowed us the freedom to carry independent research. We finally would like to thank the Basel taxpayers for providing the funding for this research and maintaining the University of Basel as a stimulating research and teaching institution.

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
