# OpenReview forum: "Designing Peer-to-Peer Systems as Closed Knowledge Commons"
_ACM.org/Middleware/Workshop/DICG — DICG 2023_

### Official Review · Reviewer_emqr · 2023-10-27
**Content producers and consumers on a P2P network**

**Rating:** 7
**Confidence:** 3

**Review:**

This paper discusses the possibility for peer-to-peer systems to support closed dissemination of a content (updates) that some peers generate.  The paper is well-written and presents an original idea. This is an overall good paper for a workshop that should generate interesting discussions.

Strengths:
* Interesting idea that challenges the reader.
* Paper is well written and despite the space limits does a good job at discussing details.

Weaknesses:
* A content dissemination system usually relies on an overlay to distribute the load among users. It seems that such an overlay is not used here, which might limit the scalability of the proposed system.
* Eventual consistency does not prevent double spending and blacklisting a user does not prevent them to return under a different identity. Would it be possible to financially punish a user that deviates from the protocol?
* P2P publish/subscribe systems could be discussed

Remarks/Typos:
* I would split the last sentence of Sec. 1 in multiple sentences. It is heavy as is.
* ‘Once they have obtain’ (Sec. 2.1): once they have obtain
* ‘We expect thie’ (Sec. 2.4)

---

### Official Review · Reviewer_N55f · 2023-10-27
**Review of Designing Peer-to-Peer Systems as Closed Knowledge Commons**

**Rating:** 4
**Confidence:** 3

**Review:**

The paper addresses the issue of distributing collectively produced append-only digital content under a proprietary model and suggests that P2P systems are more suitable for such cases compared to cloud-based systems. It introduces an economic model that involves two types of users: content providers (those who produce content) and contributors (those who pay for and use the content). The authors also present the proposed economic operations and discuss the incentives for honest participation. The paper argues that hosting this system on a P2P platform is more economically viable than using cloud-based platforms.

The paper is well-structured and presented, and the problem it addresses aligns perfectly with the workshop's theme. However, I have some concerns regarding the paper, mainly threefold:

1. The arguments presented in the paper seem weak and lack depth. Sections 2.4 and 4 constitute the main argumentation in favor of the proposed solutions. Most of the time, the arguments lack sufficient evidence to support their claims. For example, the authors provide three arguments supporting the idea that contributors are incentivized. The first argument (Section 2.4, paragraph 3) refers to Section 4, which also contains unsupported claims. The second argument, about the incentive to use an update-based system, is not directly related to the P2P nature of the system but more about how the log is structured. The third statement argues that users are responsible for "maintaining the ability of" content providers to have "sufficient livelihood," neglecting any consideration of adversarial behavior. In the same paragraph, the authors argue that "We expect this mutual dependency between the developers of the system, the content providers, and contributors will contribute to developing strong loyalty." However, in such interactions, relying solely on reciprocity can bring success only until one party diverges from collaboration. Furthermore, even if the natural dependency mentioned by the authors naturally exists, it is not necessarily an outcome of the proposed solution. This raises the question: if such a natural dependency is sufficient, meaning that everyone is honest and good, why do we even need time-based or update-based subscriptions? Everyone would honestly pay for what they consume, as they consume.
2. There appears to be a discrepancy between the problem framing in the Introduction and the details of the proposed solution. For example, in the introduction, it is stated that the addressed problem "is partly caused by the technical workings of the peer-to-peer protocols themselves: replicating data on every user's devices precludes the ability to control access to data after it has been replicated." However, this observation is neglected in the rest of the paper, as the authors later agree that "our model does not prevent some participants from freely distributing restricted updates" in Section 2.4. Another example is that the general replication problem space introduced in the Introduction is restricted to "producing knowledge in the form of updates," neglecting all cases that cannot be presented in the form of updates. Finally, in the introduction, the problem seems to be a general case, but as the paper progresses, it narrows down to the "software update logging" problem (See Section 2.4, paragraph 2).
3. There is a lack of consistency in terminology. Despite the simplified user model (only producers and contributors) presented in Section 2, the rest of the sections mention user types such as "users," "original developers," and "content providers," making the paper difficult to follow.


Additional comments:

- The role and responsibilities of relayers in the P2P network protocol are not clearly explained. The strength of P2P protocols comes from their lack of fixed roles, as anyone can be a producer or consumer at any time. It would be helpful to clarify how relay nodes fit into the proposed protocol.

- In relation to the previous comment, the authors mention in Section 2.2 that they do not compensate the use of computing resources of other contributors partly to avoid creating a resource market. However, it's not clear how they intend to incentivize relay nodes in this context.

- Section 2.3 "determining the time elapsed  since the subscription began is performed by another contributor providing updates" It is not clear who is that "another contributor"

- In Section 4, paragraph 2, it is not clear how the conclusion in the last sentence of the paragraph was reached. More context and explanation are needed.

- The conclusion should explicitly address the significance of comparing the feasibility of applying the proposed model to cloud-based systems. If the model is primarily designed for P2P systems, the paper should explain why this comparison is relevant and what insights it provides.

---

### Official Review · Reviewer_2w2p · 2023-10-31
**Novel model of digital commons, revisiting after 25-years: Contribution-Conditional Replication**

**Rating:** 7
**Confidence:** 4

**Review:**

This paper introduces a collaborative system based on The Commons paradigm, specifically Contribution-Conditional Replication.
It introduces an economic model which seems to be Internet-friendly and avoid restrictive right management or "trustworthy playback hardware". The economics of the content-creators has been unsolved since the days of Napster and Gnutella. Companies such as OnlyFans take 20% of the user donations. This paper proposes an architecture compatible with todays influencer economy in which artists obtain 100% of the donations by fans and superfans.

Not only the economic model is strong also the scientific side has merit. Their replication primitive is intended for eventually-consistent replicated databases, with git as a practical implementation. This choice gives the idea a significant body of related work and available tooling. This work is focused on an unsolved problem which is still highly relevant in todays world. It provides a concrete step forward for realisation of The Commons. Producers are required to "contribute tokens for each individual log to be replicated by the system". This means microtransactions are key to functioning of the proposed commons.

Finally, the drawbacks of this paper. Significant space is allocated for the economic model, leaving little space for original work. The "Algorithm 1" contribution on page 4 is based on decades of prior work on epidemic protocols. Even for a workshop a more substantive contribution would have been desired. The need for 2P-BFT-Log complexity is unclear, each produces issues their own exclusive token. Updates could also simply be serialised, the need for conflict resolution and update merging is unclear for current alternative to the social media landscape.